# Evaluating the effectiveness of integrating biofeedback in the treatment of aggressive outbursts (BRET-IA²): A study protocol

Alberto J. Molina-Cantero[1]*, Isabel Rojas-Pérez[2], Montserrat Gómez de Terreros-Guardiola[3], Isabel Gómez-González[1], José C. Vidosa-Batllés[4], Teresa de Jesús Bermejo-González[5], Manuel Merino-Monge[1]

1 Departamento de Tecnología Electrónica, ETS de Ingeniería Informática, Universidad de Sevilla, Seville, Spain, 2 Departamento de Psicología. Clínica Inebir, Seville, Spain, 3 Departamento de Personalidad, Evaluación y Tratamiento Psicológicos. Universidad de Sevilla, Seville, Spain, 4 OWIP Technology, Seville, Spain, 5 Departamento de Neuropediatría. Instituto Hispalense de Pediatría, Seville, Spain

☯ These authors contributed equally to this work.

* almolina@us.es

## Abstract

**Introduction:** This study provides a comprehensive overview of the materials and methods used to evaluate the effectiveness of the use of biofeedback in the treatment of aggressive episodes in children and adolescents.

**Background:** Aggressive episodes are common in various disorders and are associated with deficits in emotional processing and impulse control, primarily due to dysfunctions in the amygdala and prefrontal cortex (PFC). These brain regions also regulate physiological arousal, influencing heart rate and other autonomic functions even before aggression manifests. These early signals can be shown to the person (biofeedback) reinforcing therapeutic skills to enhance emotional regulation and reduce aggression.

**Methods:** A total of 70 participants will be recruited for a randomized controlled trial (RCT). All participants will receive therapy, although only the intervention group will incorporate biofeedback. The experimental study will be split into three blocks: (1) *Home Monitoring*: Physiological signals will be recorded using a smartwatch, and aggressive episodes will be captured with a camera; (2) *Laboratory Assessment*: Participants will attend three sessions, where therapists will induce aggressive reactions, using the video clips recorded at home. Simultaneously, real-time physiological signals will be measured. These sessions will also include relaxation periods before and after the provoked outburst; (3) *Therapeutic Intervention:* Similar to the laboratory assessment block, therapists will induce aggressive responses in three sessions; however, in this block, participants will receive therapy. Additionally, participants who belong to the intervention group, will include biofeedack in the therapy. Biofeedback is focused on heart rate (HR), heart rate variability (HRV), and skin conductance level (SCL). The CACIA, the Stroop, and other pre- and post-experimental tests. will be used to assess the differences between the control and intervention groups.

**Data availability statement:** No datasets were generated or analysed during the current study.

**Funding:** This research was funded by project Bret-IA2, Grant PID2023- 147508OB-I00, funded by MCIN/AEI/10.13039/501100011033. The funders had no role in study design, data collection and analysis, decision to publish, or preparation of the manuscript.

**Competing interests:** The authors have declared that no competing interests exist.

**Discussion:** Emotions play a fundamental role in decision-making, social interactions, and mental health. Emotional dysregulation often leads to aggression, irritability, and anxiety. Showing physiological responses to patients, such as heart rate variability and skin conductance, may improve emotional awareness and regulation. This study aims to verify the effectiveness of including biofeedback in such therapy.

## Introduction

Aggression is a common symptom that is frequently addressed by mental health professionals, yet it continues to be one of the most complex issues to treat. From a psychiatric-diagnostic perspective, aggressive behavior has been subsumed under specific diagnoses of conduct disorder (CD) and oppositional defiant disorder (ODD), although many other disorders can also affect behavior, e.g., autistic spectrum disorder (ASD), intermittent explosive disorder (IED), or attention-deficit hyperactivity disorder (ADHD) [1].

A common feature of aggressive behavior is the reduced ability to *process emotions* and *control impulsive conduct* (behavior regulation) [2–4]. Specific areas of the brain are specialized in these two main functions. Evidence points to dysfunctions in the amygdala and prefrontal brain regions as key contributors to aggressive behavior. The amygdala, in particular, shows varied responses: reduced responsiveness to fearful faces may impair the recognition of distress cues and empathy, leading to deficient control of aggressive behavior [5]. This is aligned with the violence-inhibition mechanism model [6], suggesting a lack of empathy due to impaired distress recognition. Conversely, some studies indicate hyperresponsiveness of the amygdala in certain contexts, reflecting excitement about brutal situations [7]. This suggests that the amygdala's role in aggression may differ based on the emotional context, such as anxiety-induced aggression versus emotional insensitivity [8].

Moreover, the prefrontal brain regions, including the orbitofrontal and ventromedial prefrontal cortices, as well as the dorsal anterior cingulate cortex (ACC), exhibit dysfunctions associated with aggressive behavior. These regions are crucial for regulating affective responses, and their dysfunction may result in reduced ability to restrain aggressive impulses. Specifically, the orbito-frontal cortex (OFC) is critical for evaluating the reward and punishment values of stimuli. It also contributes to the regulation of emotional responses, social behavior, and is involved in inhibiting inappropriate responses and controlling impulses [9]. The ventro medial prefrontal cortex (vmPFC) is integral to decision-making, emotion regulation, and social cognition [10]. Its extensive connections with various brain regions (such as the amygdala, dorso-lateral PFC, hypothalamus, ACC, etc.) facilitate these complex functions. The vmPFC is strongly connected to the ACC, which is involved in emotion regulation, error detection, and reward-based learning. Abnormalities in these areas have also been observed during reinforcement processing, indicating deficient learning mechanisms related to social rules and appropriate behavior.

Additionally, the capacity of correctly perceiving body states (interoceptive accuracy), relates to interpersonal problems, including aggression. Interoption refers to the process by which the central nervous system (CNS) receives and integrates information about the internal state of the body. The parasympathetic nervous system (PNS) and sympathetic nervous system (SNS) are also two major ascending pathways to transmit interoceptive signals to the CNS. Interoceptive signals are firstly processes in subcortical areas of the brain and then projected to higher brain regions including hypothalamus, insula, ACC and somatosensory cortex for integration [11]. Firstly, primary interoceptive information is relayed from thalamus

to insula where the integration with exteroceptive signals is performed. The insula, especially, the anterior cortex is strongly connected to the OFC, ACC and PFC, which may be involved in linking interoceptive with emotional or cognitive states, and in generating regulatory signals that are sent back to lower brain regions associated with interoceptive descending efferent systems.

Difficulties in self-regulation (which includes inhibitory control) linked to interoception can contribute to these problems in individuals with borderline personality disorder (BPD) symptoms [12]. Our internal emotional state shapes our perceptions of others. During a stressful interview, the researchers found that participants with lower interoceptive accuracy were more likely to misattribute their own stress responses to negative traits in the interviewers, perceiving them as less helpful or more hostile. In contrast, individuals with higher interoceptive awareness were better able to distinguish their internal stress from external cues, leading to more accurate and unbiased social judgments. These findings highlight the role of interoception as a protective factor in reducing social misperceptions during emotionally charged situations [13].

The amygdala and PFC, along with many other brain regions, are connected to the hypothalamus [14], which regulates the sympathetic and parasympathetic branches of the autonomic nervous system (ANS) that innervate the heart, sweat glands, and other organs. As a result, the activation or deactivation of these areas induces changes in heart rate, blood pressure, skin temperature, respiration rate, and sweating, leading to a certain level of physiological arousal. Consequently, the detection of peripheral activity could help patients identify their emotions. For example, it is known that heart rate is accelerated or reduced by the activity of the SNS and PNS, and, for this reason, it can be associated with emotional dysregulation [15]. Some authors have monitored heart rate and its variability concerning aggressive episodes using off-the-shelf wearables [16]. Elevated heart rates can occur during an outburst, although authors did not find conclusive results due to the population size used (3 participants). Another study has also reported significant differences between diminished HRV in people with IED, depression, and post traumatic stress disorder (PTSD) when compared to healthy volunteers [17]. Similarly, other studies focused on skin sweating, such as that of Tonacci et al. [18] observed higher levels of sympathetic arousal in SCL for anger and sadness stimuli in children with aggression problems, and Deutsch et al. [19] detected an increase in skin conductance in subjects with outbursts of combative behavior.

Aggressive behavior is frequent during childhood and adolescence, which can significantly disrupt the functioning of both the child and their family. The onset of aggressive behavior at earlier ages is linked to higher biological risks and often results in life-course persistent antisocial behavior. A better long-term prognosis, leading to a relatively normal daily life, is more likely when aggressive behavior emerges at older ages [20]. For this reason, it is important to apply treatment at an early age. There are two types of non-exclusive treatments: drug administration and psychological therapy. The former addresses symptoms by altering the biochemistry of the patient's brain, thereby reducing the expression of the disorder. However, no pharmacological treatment has yet targeted the core symptoms of disorders such as ASD, which include social and communication deficits, repetitive behaviors, restricted interests, and abnormal sensory processing [21].

We focus on equipping individuals with the necessary skills to manage their behavior through psychological therapy. This therapy is enhanced by using technological devices that help identify internal states through the use of physiological signals (biofeedback), like HRV, that enchance interoception, and allow warning individuals of an imminent aggressive outburst. Biofeedback is a well-known technique that has been tested in several areas, for example, to improve attention in children with ADHD through a game which an avatar that has to

reach a target by modulating the beta activity obtained from frontal areas [22,23], or in the treatment of depression [24].

It has been shown that neuronal activity can be modulated by peripheral biosignals. In [25] it was observed that respiratory rate, both spontaneous and voluntarily controlled, decreased alpha band power in phases in inspiration and increased with expiration in sensorimotor areas. The use of biofeedback based on SCL has shown an effective decrease in anxiety, with a significant reduction in sympathetic activation and sympathetic balance [26]. Likewise, the use of HRV in a biofeedback system showed that an increase in this parameter in the resting state decreased negative emotions [27], and proven useful in in regulating stress [28]. Other biofeedback methods have included the use of electromyography (EMG) to allow managing pain conditions sucessfully [29].

The main goal of this study is to verify the efficacy of psychological treatment, in the young population, when this treatment includes biofeedback of changes in some physiological features (such as SCL, HR and HRV) associated with the imminent onset of an aggressive outburst. Apart from the main goal, other aspects will also be explored: how aggressive episodes influence physiological signals (HR, HRV, SCL in particular) identifying the most significant features; and how different scenarios (home, lab) influence this variability in physiological signals.

## Materials and methods

### Participants

As explained above, the early onset of aggressive behavior is linked to higher biological risks and often leads to persistent antisocial behavior, highlighting the importance of applying treatment at early age. For this reason, the group to be studied will consist of 70 participants aged 10–16 years. They will be recruited from the INEBIR clinic, where the experimental sessions will be carried out, and other participants will be recruited from the Instituto Hispalense de Pediatría. The sample size was by analyzing the differences of the means according to a Student's t with a confidence level of $\alpha = 0.05$, a test power of $1 - \beta = 0.8$ and a moderate effect size [30]. A moderate effect size (Cohen's d = 0.5) represents a practically meaningful difference in the context of clinical decision-making, even if smaller effects may also be statistically detectable in larger samples and it is a commonly used convention in behavioral and clinical research when planning exploratory studies [31]. A moderate effect size results in 62 participants. For a small effect size (Cohen's d = 0.2), we would need to recruit more than 300 participants, and we do not have the human and material resources to complete this study in a relatively short time. Additionally, to account for potential non-response or dropout, we included an additional 10%, bringing the final target sample size to 70 participants. This approach ensures sufficient power to detect differences between groups while anticipating common challenges in recruitment and retention.

To participate in this study, the following inclusion criteria will be applied:

- Adolescents aged 10 to 16 whose parents seek professional help because their son or daughter exhibits aggressive outbursts that they are unable to manage.
- Intermittent aggressive episodes with a frequency of at least once a week in the two months prior to the start of the intervention.
- Not presenting some of the following:
  - History of bipolar or psychotic disorder.
  - Head trauma with loss of consciousness for more than 60 minutes.

Participants will be randomly divided into two groups: a control group (children who will not receive the experimental treatment based on biofeedback), and an intervention group, with an equal number of participants in each group. To avoid bias and to have the groups as balanced as possible, the allocation will be done in two stages. In the first, the subjects are divided by age range and biological sex, so that four subgroups are created, male between 10–13 years and between 14–16 years, and female between 10–13 years and between 14–16 years. Finally, in the second phase, individuals from each subgroup are randomly assigned to the control or intervention group, using a computer-generated sequence of random numbers and maintaining a balance between the groups, ensuring unbiased distribution. Participants' names will be blinded by the recruiting staff from the INEBIR setting.

**Tentative participants' recruitment dates.**   Recruitment will start in September 2025, and it is expected to take approximately one month. Data collection and results are anticipated to be completed by March 2026 and July 2026 respectively.

## Sessions

Participants who belong to the intervention group will perform a total of eight one-hour-a-week laboratory sessions (S2–S9), while the control group will attend seven sessions (S2–S4, S6–S9). The intervention group has an additional session, S5, to help participants identify the physiological values before aggressive outburst through biofeedback. Both groups will have in common an informative and strategy selection session (S0) and one at-home session (S1) to capture some biosignals and record videos that include aggressive episodes. Figs 1 and 2 show the chronological order of the blocks and sessions in the experiment.

**Block 1 or initial block.**   It consists of two sessions:

S0   In the first session, parents and adolescents will be informed about the procedure of the experiment and asked for cooperation. Those who agree to participate must sign the Informed Consent document. In this session, participants fulfilling the inclusion criteria will undergo the  first psychological assessment consisting of the following tests:

- CACIA [32]. This questionnaire evaluates self-control by means of four scales: three positive (Personal Feedback, Reward Delay and Criterial Self-Control) and one negative (Processual Self-Control). It also incorporates a Sincerity scale. Its scales focus on the evaluation of Self-Control considered from a behavioral point of view, whose basis is the conscious effort of the person to modify his or her reactions.
- STROOP test [33]. This reference test is employed in the detection of neuropsychological problems and brain damage. It allows evaluating the phenomenon of interference, which is strongly linked to inhibitory control processes.
- Wisconsin Card Sorting Test (WCST) [34]. This test assesses the ability required to develop and maintain the problem-solving strategies necessary to achieve a goal. It is particularly sensitive to lesions involving the frontal lobes (Stuss et al., 2000), thus it is very useful for discriminating between frontal and non-frontal lesions.
- The Child Behavior Checklist (CBCL) [35] is a standardized assessment tool used to identify emotional and behavioral problems in children and adolescents. Parents or caregivers, in addition to the children themselves (Youth Self Report (YSR)), complete the checklist by rating their behavior, providing valuable insights into various domains such as anxiety, depression, and aggressive behavior. The CBCL and the YSR are part of the Achenbach System of Empirically Based Assessment  (ASEBA), which is widely used in both clinical and research settings to inform diagnosis and treatment planning.

|  | Enrolment | Allocation | Post-allocation | | | | | | | | | Close-out |
|---|---|---|---|---|---|---|---|---|---|---|---|---|
| TIMEPOINT | $-t_1$ | $S_0$ / $0$ | $S_1$ / $t_1$ | $S_2$ / $t_2$ | $S_3$ / $t_3$ | $S_4$ / $t_4$ | $S_5$ / $t_5$ | $S_6$ / $t_6$ | $S_7$ / $t_7$ | $S_8$ / $t_8$ | $S_9$ / $t_9$ | $t_{10}$ |
| **ENROLLMENT:** | | | | | | | | | | | | |
| Eligibility screen | X | | | | | | | | | | | |
| Informed consent | X | | | | | | | | | | | |
| Allocation | | X | | | | | | | | | | |
| **INTERVENTIONS:** | | | | | | | | | | | | |
| Recording at home | | | X | | | | | | | | | |
| Evoking aggressive outburst | | | | ●——————● | | | | | | | | |
| Using Biofeedback to recognize internal variables (HR, HRV and SCL) | | | | | | | X | | | | | |
| Applying behavior and emotional self-regulation therapies | | | | | | | | ●—————————● | | | | |
| **ASSESSMENTS:** | | | | | | | | | | | | |
| Frequency of aggressive episodes, and several tests: CACIA, Stroop, WCST, CBCL, YSR | | X | | | | | | | | | | X |
| Recording physiological signals with the smart watch at home | | | | X | | | | | | | | |
| Recording HR, HRV, and SCL at laboratory | | | | ●——————● | | | | | | | | |
| Use of HR, HRV and SCL at laboratory in intervention groups | | | | | | | | ●—————————● | | | | |

**Fig 1.** Schedule of enrollment, interventions, and assessments in accordance with Spirit recommendation.

S1  Families will be the active part of the experiment in the second session, which takes place at home, not at the research center. A small camera will be hidden in the living room, while the adolescents wear a wristband that records multiple biosignals (HR, SCL and skin temperature (ST)). The goal is to register the scene together with physiological variables before, during and after at least up to four aggressive outbursts.

The experimenter will synchronize both the camera and the wristband, and instruct parents to extract the physiological information, make the video clip containing the aggressive episodes, and annotate when they started.

**Block 2.** The second block consists of a recording session at the laboratory, which is repeated three times.

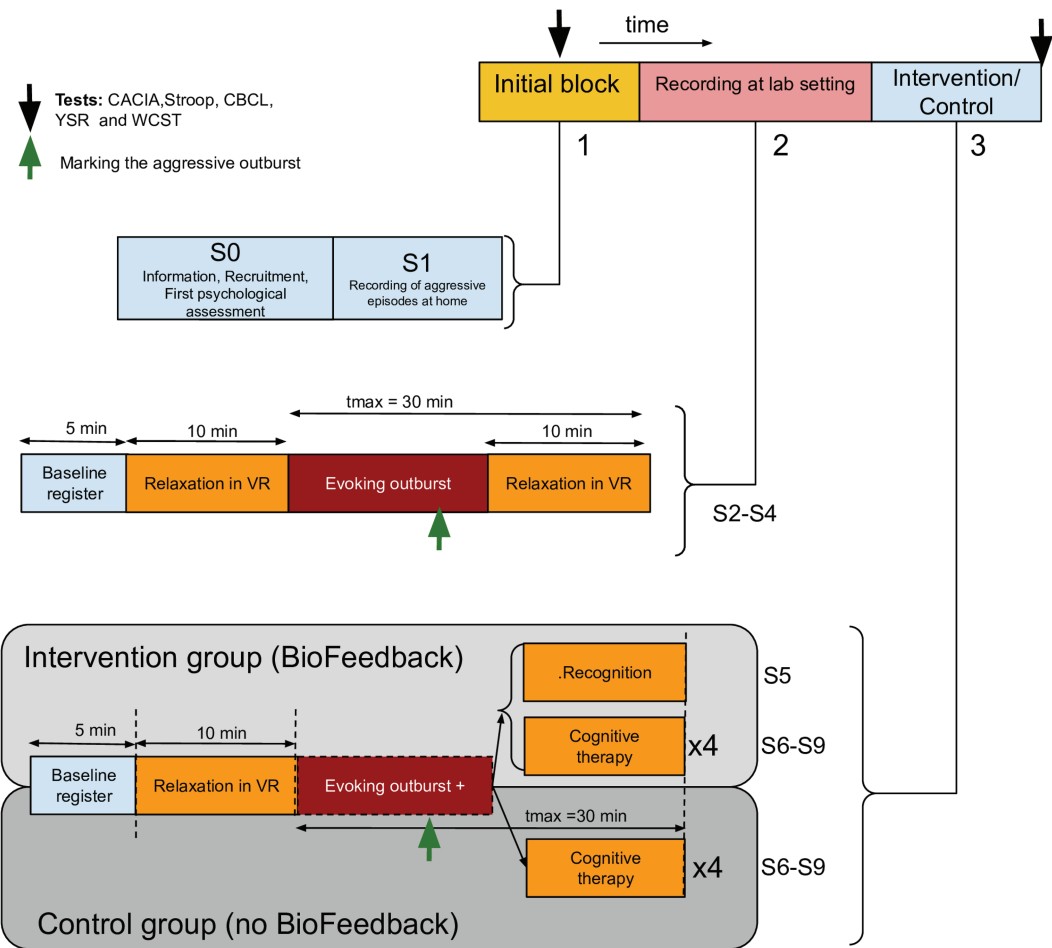

**Fig 2. Experimental session schedule. The experiment consists of three main blocks.** 1) Initial Block, which includes two sessions: Session 0 (S0) (providing information to parents and the recruitment of participants); and Session 1 (S1) (recording at home using a smartwatch and a spy camera). 2) Second Block, which includes three sessions (S2-S4) where physiological signals are recorded before, during, and after an aggressive event. Each session consists of the following phases: baseline recording; relaxation using virtual reality (VR); evocation of the aggressive outburst by watching a prerecorded video clip; and relaxation. 3) Final Block: The subsequent sessions follow a similar phase arrangement as the second block, except for the last part. For intervention group participants, this phase will be replaced with cognitive therapy (S6-S9) with biofeedback and a session dedicated only for recognition of biofeedback signals (S5). For controls, the last phase will be replaced with cognitive therapy (S6-S9) without biofeedback.

S2–S4 In these sessions, parents, scientists, adolescents and psychologists will meet to try and reproduce the outburst and record physiological signals before, during and after the aggressive episode.

These sessions contain several phases:

- **Baseline register.** Initially, the researchers will set up the system and start recording a baseline reference for approximately 5 minutes [36].
- **Relaxation.** After the baseline register phase, adolescents will move on a relaxation phase, in which they will use an immersive Virtual reality (VR) system for 10 minutes. In this phase, participants can move into a landscape while listening to ambient sounds (birds and sea waves). VR, particularly immersive virtual reality (IVR), has demonstrated its effectiveness in therapy with people with behavioral disorders,

improving emotion recognition, social skills, and attention, while diminishing their anxiety and phobias [37–39].

- **Evoking an outburst.** In the following phase, adolescents will watch the recorded aggressive outburst while their parents explain to the therapist the reasons that caused such an episode, including all the necessary details that favor the elicitation of a new outburst. Different videos will be used in each session to avoid the accommodation effect. Re-experiencing a situation that previously triggered an anger outburst can indeed provoke it again. This is because recalling emotionally charged events can activate the same brain areas involved in the original emotional experience [40]. According to the classical conditional model, if a particular situation has been previously associated with an anger response, re-exposure to that situation can trigger anger again due to conditioned memory [41].
- **Relaxation.** At the end of the episode, parents will leave the room for a while, and the participants will go to a new relaxation phase for another 10-minute period using again the VR.

Fig 2 depicts the phases and actions in this important session. The experimenter will tag the beginning of each phase in this session and will mark the moment when the impulsive outburst appears. By doing so, researchers will be able to analyze offline how the physiological information changed and how it can be included in the therapy. There will be a digital diary to collect tests outcomes and any relevant information regarding the patient during the session, including whether the outburst was evoked successfully. Data will be discarded from analysis when it is not possible to evoke the outburst.

The data collected in this block will allow assessing the influence of an aggressive episode on physiological signals. This will be used to teach individuals to identify when an outburst is about to occur, and prevent it.

**Block 3.**

S5 Only for the intervention group. The adolescents will be trained to identify the physiological response when they are relaxed and when an outburst is coming. It is possible to use these responses to predict the aggressive outburst at least one minute before it manifests [42]. To do this end, the VR system receives and shows HR, HRV, and SCL on the virtual scenario. Therapists will help participants identify the physiological features related to aggressive outburst based on the data collected in previous sessions (Block 2).

S6–S9 They are similar in both intervention and control groups, and follow the same layout as the sessions in block 2, excluding the last phase:

- **Cognitive therapy in the control group.** After evoking an outburst, adolescents will be trained with cognitive, behavioral and emotional self-regulation therapies [43], which have proven their effectiveness for managing anger and learning positive coping skills. The underlying theory is that people can minimize their negative feelings and behaviors when they are aware of their irrational beliefs and work to change their minds, by focusing on them continuously. With this therapy, people are endowed with strategies to improve their capabilities of identifying negative thoughts and beliefs in situations of conflict and/or frustration, which are the main triggers of aggressive outbursts. The goal is to ensure that, through identifying such thoughts and becoming more aware of their emotions, the individual gradually dismantles those irrational thoughts.
- **Cognitive therapy in the intervention group.** Participants will follow the same cognitive therapy as the controls, although they will receive information through

biofeedback. By identifying this information and applying the skills learned in therapy, participants are expected to better manage or avoid imminent outbursts.

At the end of session S9, the CACIA, Stroop, CBCL, YSR and WCST tests will be assessed again. The differences between the new obtained physcological scores, with respect to the initial scores, along with the differences between control and intervention groups, will be used to assess the efficacy of the intervention.

## Materials

**Wifi camera.**   A camera (model Tapo C200), which captures video and audio, will be hidden in the living room, which captures an aggressive outburst. The experimenter will synchronize this camera to the Empatica E4 wristband.

**Empatica embrace plus.**   Adolescents will have to wear this wristband for several days at home. It records multiple biosignals (photoplethysmogram (PPG) [44], electrodermal activity (EDA) and skin temperature (ST)) at a sampling rate of 64Hz. Heart rate (HR) and heart rate variability (HRV) can be extracted from photoplethysmogram (PPG) and electrodermal activity (EDA) allows estimating the SCL. The camera will be synchronized to the wristband using their own internal clocks.

**Physiological recording system.**   For the sake or reproducibility of the laboratory sessions, several devices, not mutually exclusive, can be used to capture subject's biosignals. Some of them are shown below:

**AAI Wearables.** The wearable devices described in [45] capture EDA and electrocardiography (ECG) and transmit raw data through Bluetooth low energy to a laptop that converts the incoming data into LSL streams. LSL (Lab-Streaming Layer) is a C++ library that allows data synchronization from multiple sources [46].

**Emotibit.** Emotibit records the same physiological signals as Empatica E4 [47]. It is less compact but more affordable. It also allows real-time signal transmission, although its battery life is more limited. As in the case of Empatica, Emotibit will be placed at the wristband during the laboratory sessions. An application called Emotibit Oscilloscope, running on a laptop, can receive, through WiFi connection, data from the Emotibit wristband, and then redirect the physiological signals to LSL streams on the computer network.

**Psychobit** BITalino PsychoBIT [48] is a pre-assembled device designed for basic psychophysiological data acquisition. It facilitates the measurement of various physiological signals, including pulse, blood pressure, respiration, and skin conductivity. The core of PsychoBIT is BITalino (r)evolution Plugged, which offers Bluetooth communication for seamless data transmission. The kit comprises several pre-assembled sensors: ECG, EDA, PPG, etc. Data are received by OpenSignals, which is a free software that allows redirecting raw or processed information, such as HR, which is obtained from ECG or PPG, to LSL streams.

Fig 3 shows a possible schematic based on EmotiBit.

**Recorder.**   The laptop also hosts the LSLRec [49], which is based on Lab-Streaming Layer https://github.com/sccn/labstreaminglayer, a C++ library that allows data synchronization. LSLRec is an easy-to-use, open-source, multi-platform, recording system developed on Java. LSLRec reads data from LSL streams and stores them while maintaining synchronization with the experimental blocks and phases of the experiment.

**Virtual reality (VR) platform.**   For the immersive VR we use the Meta Quest 3 headset, although any other commercial model of the same manufacturer could be used. The use of this device is allowed for the range of ages of the participants. The program has been developed using Unreal Engine, which is one of the most powerful real-time 3D creation tools,

Meta Quest 3 headset

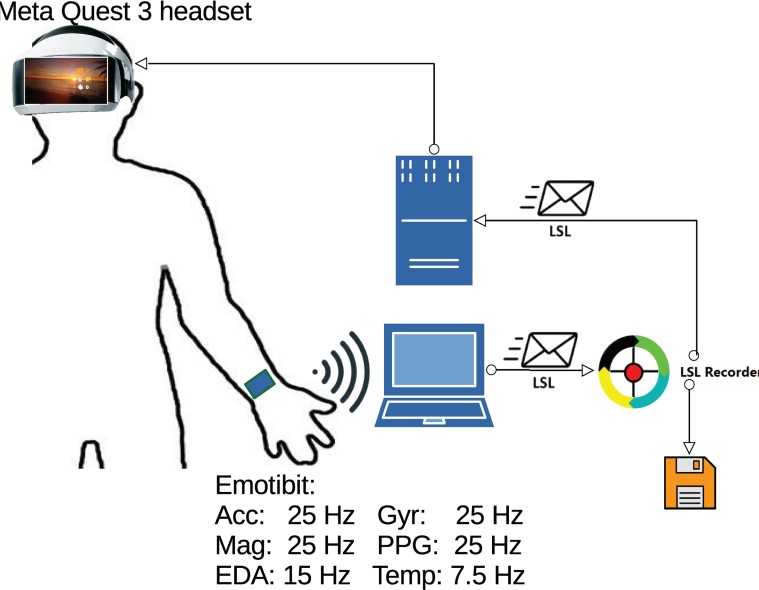

**Fig 3. Experimental setup at the clinical setting.** On the left, the person with the Emotibit for measuring physiological signals that are transmitted to the laptop. The subject also wears the VR glasses controlled by another computer that also receives physiological signals for biofeedback purposes.

running on a personal computer that features a CPU AMD Ryzen 9 3950X, with a GPU model NVIDIA GeForce RTX 3070 and a X570 AORUS ULTRA motherboard with 64 GB RAM. The computer is connected to a local network in order to receive LSL streams [46] that contain the physiological information to show on the screen.

Fig 4 shows the two scenarios included in the VR for the relaxation phase. Participants can navigate through those scenarios using only one controller, which can be configured for left- or right-handed people. In the figures biofeedback information is also shown including HR, HRV and SCL.

## Metrics

As a general procedure, outliers will be identified by applying the interquartile range method. We will then examine each outlier assess whether it could be attributed to measurements errors, technical issues, or specific characteristics of the participant. Data points will be excluded only when a clear justification exist (e.g: confirmed recording errors).

Continuous variables will be summarized using the mean and standard deviation (SD) when data are normally distributed. For variables that do not meet the normality assumption, we will report the median and interquartile range (IQR). Normality will be assessed through the Shapiro–Wilk test. Appropriate statistical tests (parametric or non-parametric) will be selected accordingly to the distribution of residuals.

**Psychological assessment.** Scores will be obtained by CACIA, STROOP, WCST, CBCL, and YSR tests at the beginning and at the end of the experiment. These tests contain 40 variables that will help assess the effect of the therapy and look into the underlying cognitive mechanisms regarding aggression, such as executive functions and other emotional or behavioral issues associated with it.

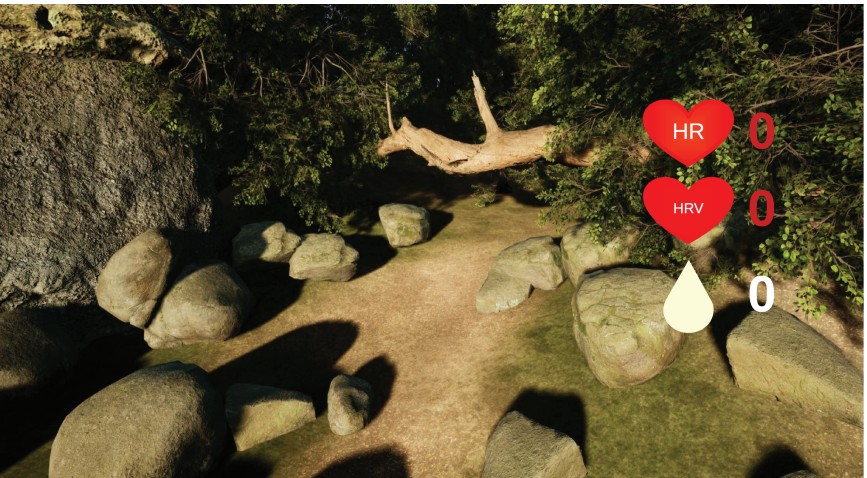

**Fig 4. VR scenario.** One of the VR scenes showing some physiological variables for biofeedback.

At the end of this section reader will find a list of the primary variables that directly measure the impact of the therapy and reflect key aspects related to self-regulation/-control, which are important to mitigate outbursts.

**Cardiac features.** One important feature, from which others can be derived, is instantaneous heart rate (HR). Both Empatica and Emotibit obtain HR from PPG signals by determining the position of their maximums and then calculating the time difference between two consecutive maximums.

Derived from HR, heart rate variability (HRV) in time analysis can be extracted from 2-minute windows by obtaining the root medium square (RMSSD) between maximums of the PPG peaks. The length of the windows will be selected to fulfill the requirements of frequency band analysis of the heart rate variability (HRV) spectrum [50].

HRV is especially interesting as it allows assessing the activity of the parasympathetic and sympathetic pathways of the autonomic nervous system (ANS) [51]. For instance, the ratio between low frequency, [0.04–0.15] Hz (LF) and high frequency, [0.15–0.4] Hz (HF), which is called the ratio between LF and HF (LF/HF), shows the balance between the SNS and the PNS.

HR and HRV will be used for biofeedback, although a much richer feature set will also be included for offline analysis [52].

**Skin conductance level (SCL) measurements.** The variation of the electrical conductance is referred to as EDA which is primarily due to sweat gland secretion. This activity is closely linked to the sympathetic nervous system, which is responsible for the body's "fight or flight" response and therefore used as an indicator of physiological arousal, emotional responses, and stress. The two main components derived from EDA are skin conductance response (SCR) and skin conductance level (SCL).

SCL is the parameter that will be included in biofeedback and represents the baseline level of skin conductance over time, linked to emotional arousal.

**Summary of primary variables.**

- Phsycometric measures
  - **CACIA**:
    - **Delay of gratification percentile (RR)**: This percentile indicates the ability to postpone immediate rewards for larger, later benefits, a key aspect of impulsivity and self-control. The higher percentile the better self-control in delaying gratification.
    - **Criterial Self-Control Percentile (CSC)** : This percentile reflects a general capacity for self-control in situations requiring judgment and adaptation to specific criteria
  - **CBCL and YSR**
    - **Aggressive behavior T-score**: A continuous variable reflecting the severity of aggressive behaviors reported by parents (CBCL) or the youth themselves (YSR). Scores above 60 or 65 are often considered in the clinical range.
    - **Externalizing problems T-score**: This variable asessess a broader spectrum of outward-directed problem behaviors, including aggression, rule-breaking, and defiance. Higher scores indicate more severe externalizing problems.
- Physiological measures
  - **HR**: Measured in bpm, this continuous variable is affected by both PNS and SNS.
  - **HRV (RMSSD)**: A continuous variable that also shows the activity of the ANS. A reduced HRV is associated to a high SNS activity in healthy subjects.
  - **SCL**: This is a continuous variable, measured in microsiemens ($\mu S$), that reflects sympathetic activity linked to emotional arousal.

## Analysis

The statistical analysis will be performed in Matlab 2024.

**Assessment of cognitive therapy with biofeedback.** The main goal of this study is to verify the efficacy of the proposed therapy. At the beginning adolescents will fill in the CACIA, Stroop, WCST and CBCL tests. At the end of the therapy, these metrics will be assessed once more to compute the difference with the initial tests. These differences will be compared between the intervention and control groups to ascertain statistical significance. To this end, we will use the Mann-Whitney-Wilcoxon test, or the one-way ANOVA, depending on the distribution of residuals, to determine whether samples originate from the same distribution

**Significant physiological features for aggressive outbursts detection.** We will also extend the statistical analysis to gauge how an aggressive outburst influences all physiological signals. To attain this objective, we will collect features from the last minute of relaxation period, in order to build a set to compare with the features associated with the 60-second data segment that contains the aggressive episode. For each participant there will be a maximum of 7 sets of features for any physiological signal, which will be obtained from sessions S2–S4 and S6–S9, considering that the evoking phase is successful. Statistical analysis will be applied to every feature individually. Friedmann's test or ANOVA, with repeated measures will be applied to assess the difference between measures before and during the outburst. Friedmann's test is a non-parametric method suitable for analyzing ordinal data or data that do not meet the assumptions of normality and homoscedasticity; it is also specifically designed for repeated measures or matched group data, allowing for comparisons across multiple conditions or time points for the same subjects. ANOVA will be applied for normally distributed data.

The analysis will be applied to each participant individually and for all participants globally. In order to reduce biases among individuals, features obtained during the baseline phase will be removed from the other phases.

**Comparison between lab and home scenarios.** Friedman's test will verify the statistical significance of the SCL and cardiac features between the two scenarios: home and clinic. Firstly, researchers will label the time point in which an aggressive episode takes place in the temporal series. The 60s data after those labels will be segmented using the window lengths and overlaps of 30s and 50% respectively. Features will be obtained for each window and grouped into two sets, which will be used as input to Friedman's test or to ANOVA with repeated measures.

**Data missing.** A detailed strategy for handling missing data will be applied in this study, where two independent groups (control and intervention) are compared.

- For psychometric variables:
  If missing data is less than 5%, listwise deletion will be applied, excluding participants with incomplete data for the relevant variable.
  If missing data is 5% or more, Multiple Imputation (MI) will be used. This involves generating multiple complete datasets by predicting missing values based on observed variables (including group assignment and auxiliary variables) and then combining the results using Rubin's Rules [53].
- For physiological variables: Linear interpolation will be used to estimate missing values between preceding and succeeding observed time points. This method is considered suitable for smooth, continuous time-series data.

Across all variable types and methods, a sensitivity analysis will be conducted. This involves comparing the main study conclusions under different assumptions about missing data or using alternative handling methods (e.g., "worst-case" imputation). The results of this analysis will be fully reported to assess the robustness of the findings and provide transparent interpretation of any limitations.

## Ethical, legal, and security aspects

The researchers of the project agree on adhering to the ethical standards and requirements of good clinical practice. At all times, the current regulations on the development of research will be respected, as well as what is contemplated in the Declaration of Helsinki of the World Medical Association and the Oviedo Convention of the Council of Europe, and in relation to the confidentiality of the information of the study participants.

The experiment has the approval of the Ethics Committee of the University of Seville with code **1075-M1-24** (S1 File, and S2 File in the supporting information section). All participants will receive the corresponding training sheet, along with the authorization or informed consent document.

**Applicable legislation.** Law 14/2007, of July 3, on Biomedical Research. "Standards of good clinical practice" and ethical principles for research with human beings established in the "Declaration of Helsinki of the World Medical Association", revised in October 2013. Council of Europe Convention for the protection of human rights and dignity of the human being with respect to the applications of biology and medicine (Oviedo Convention). April 1997. Law 3/2001, of May 28, regulating informed consent and the clinical history of patients. Law 3/2005, of March 7, modifying Law 3/2001, of May 28, regulating informed consent and the clinical history of patients. Law 41/2002, of November 14, regulating basic patient autonomy and rights and obligations regarding information and clinical documentation.

**Steps to ensure standards of good practice.** The study establishes some steps to guarantee the rights of the participants. To achieve this, the following points will be taken into account:

- Presentation of the research project to all possible participants and their legal tutors.
- Participation is completely voluntary and requires to sign an informed consent document.
- The confidentiality of the data will be guaranteed. The researchers participating in this protocol will undertake to the confidentiality rules established by the institutions they belong to. The following laws are applicable: Organic Law 3/2018 of December 5, on the Protection of Personal Data, which guarantees digital rights; Regulation (EU) 2016/679 of the European Parliament, April 27, 2016 regarding the processing of personal data and the free circulation of this information.
- Personal information will be coded and managed through the RedCAP Database, which is a secure web application for building and managing online surveys and databases [54]. Only researchers who take part in this experiment will have access to the data.

**Patient's safety.** Users are not expected to be harmed during the experiment.

## Discussion

Emotions guide our decisions, facilitate responses to the environment and constitute the foundation of social interaction. Emotions that are appropriately expressed in their context in terms of timing and intensity are more likely to facilitate adaptive responses. Therefore, it is vital for people to be able to regulate their emotional response, in order to favor social interaction and maintain mental health. Some psychological disorders, such as ASD, IED, among many others, are characterized by emotional dysregulation. Consequently, people who suffer from them may show aggressive behavior, irritability, and anxiety.

In this context, interoception emerges as a key component in emotional self-regulating mechanisms involved in acting out [55]. Several reports have proven that a low interoception accuracy is associated with a lower emotional awareness, higher reactivity and difficulties in modulating responses under stress or frustration [56]. As the brain fails in identifying body sensations linked to emotional states (e.g: heart rate, muscular distress, etc.), a crucial information source to anticipate and regulate intense emotional responses is lost. This disconnection between body and mind may allow the appearance of acting out, due to a fail in emotional processing.

Of course, aggressive outbursts do not only appear in IED, ADHD, or ASD, as there are many personal circumstances, experiences, or psychological disorders that make people manifest aggressive episodes. We find cases in people who have suffered from a history of childhood abuse, alcohol use disorder (AUD) [57], military veterans PTSD [58], borderline personality disorder (BPD) [59], etc.

Dysfunction in the amygdala-OFC cortex network produces emotional dysregulation. Individuals with IED show irregularities within this network, with increased activity in the amygdala, a hypoactivated OFC, and reduced coupling between those areas compared to healthy people in response to social threat signals [60]. As the amygdala is important for processing emotions, and the OFC is implicated in decision-making and processing of reward/punishment, this activity pattern suggests that these people are vulnerable to intense negative emotions, have a deficit in their ability to regulate and control them, and are prone to impulsivity [61] in IED. Similarly, patients with BPD show reduced lateral prefrontal activity during emotional action control compared to healthy volunteers. Furthermore, anger has

been negatively related to prefrontal activity, while it has been positively related to amygdala activity. Therefore, deficits in lateral prefrontal cortex seem to be a common neural mechanism underlying anger-related aggression for this population [59]. This prefrontal cortex dysfunction may stem from incorrect signaling between different neural hubs. In fact, neurobiological research has demonstrated that impulsive aggression is caused by serotonin dysregulation in receptor-specific pathways in the prefrontal cortex [62].

Although emotions originate in the brain [63], their expression affects the entire human body. As explained above, physiological arousal is manifested by an increase/decrease in the activity of several organs innervated by the branches of the peripheral nervous system. For example, HR can be accelerated/de-accelerated by an increase in the activity of the SNS or PNS, respectively. Additionally, an increase in SNS activity facilitates the production of sweat in the skin [64], or hormones, such as cortisol, which increases blood pressure. This cascade of physiological responses as a consequence of an emotion can be measured by devices that show the body's reaction to people who have difficulty interpreting their emotions. This study suggests the use of primarily HR, HRV and SCL measures for biofeedback purposes, as a way of interoception enhancement. Biofeedback might enable alternative pathways in the brain to allow interpreting and integrating emotional information, increasing activity and connectivity in the insula, a region associated with internal bodily awareness, and in the ACC and PFC, which are critical for executive functions and impulse control. This improved neural regulation may lead to increased emotional awareness and a greater ability to modulate automatic aggressive responses.

## Study limitations

The application period of biofeedback therapy, along with the time required to recognize the physiological variables used to identify the preparatory state that precedes an aggressive outburst, might be too short to achieve significant results to validate the therapy. Should it be the case that the results do not show changes in the subjects, sessions S6–S9 will be repeated.

If an aggressive impulse involves numerous movements by the individual, the capture of physiological signals, and consequently the extraction of features derived from those signals, could be significantly disrupted, which in turn may affect the detection of the preparatory state.

There may also be an environmental inhibition effect when attempting to provoke an impulsive response in the laboratory. This could be a serious limitation, as it might significantly reduce the amount of valid data available for training artificial intelligence models, thereby diminishing the power of the proposed statistical analysis.

The equipment used to measure physiological signals could be damaged during aggressive responses. Although replacement materials have been planned, they might be insufficient, requiring additional budget and time to conduct the tests.

## Conclusion

By using biofeedback with cognitive, behavioral and emotional self-regulation therapies, we aim to improve the social skills of people with emotional dysregulation and reduce the frequency of aggressive episodes. This randomized control trial seeks to evaluate the effectiveness of including biofeedback in the therapy.

Although we will use HR, HRV and SCL for biofeedback purposes, a wider set of physiological features will also be studied. The goal is to pinpoint a small and significant set of features that are the best predictors for an incoming outburst.

This reduced set of features could allow designers to develop a minimal piece of equipment. The signals in the ensuing set could then be merged into a unique warning signal before the outburst so that people can apply the learned self-regulation mechanisms, and avoid or minimize the aggressive episodes.

## Supporting information

**S1 Checklist. Spirit statement checklist.**
(PDF)

**S1 File. Ethics committee approval letter translated into English.**
(PDF)

**S2 File. Approved protocol.**
(PDF)

## Acknowledgements

The authors would like to thank the participants' parents for their cooperation in this study and the INEBIR center for allowing us to use several rooms where we will deploy the materials needed for this research.

Additionally, the authors would like to thank Alejandro Gallardo-Soto, and Francisco J. Muñoz-Alonso for their support in this project.

## Author contributions

**Conceptualization:** Alberto J. Molina-Cantero, Isabel Rojas-Pérez, Montserrat Gómez de Terreros-Guardiola, Teresa de Jesús Bermejo-González, Manuel Merino-Monge.

**Data curation:** Manuel Merino-Monge.

**Formal analysis:** Alberto J. Molina-Cantero, Montserrat Gómez de Terreros-Guardiola, Isabel Gómez-González, Manuel Merino-Monge.

**Funding acquisition:** Alberto J. Molina-Cantero, Isabel Gómez-González, Manuel Merino-Monge.

**Investigation:** Alberto J. Molina-Cantero, Isabel Rojas-Pérez, Montserrat Gómez de Terreros-Guardiola, Isabel Gómez-González, Teresa de Jesús Bermejo-González, Manuel Merino-Monge.

**Methodology:** Alberto J. Molina-Cantero, Isabel Rojas-Pérez, Montserrat Gómez de Terreros-Guardiola, Isabel Gómez-González, Teresa de Jesús Bermejo-González, Manuel Merino-Monge.

**Resources:** José C. Vidosa-Batllés, Manuel Merino-Monge.

**Software:** José C. Vidosa-Batllés, Manuel Merino-Monge.

**Supervision:** Alberto J. Molina-Cantero, Montserrat Gómez de Terreros-Guardiola, Isabel Gómez-González, Teresa de Jesús Bermejo-González, Manuel Merino-Monge.

**Validation:** Teresa de Jesús Bermejo-González.

**Visualization:** Teresa de Jesús Bermejo-González, Manuel Merino-Monge.

**Writing – original draft:** Alberto J. Molina-Cantero.

**Writing – review & editing:** Alberto J. Molina-Cantero, Montserrat Gómez de Terreros-Guardiola, Manuel Merino-Monge.

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
