## [Decision Letter · Decision Letter 0]

22 Jun 2025

PONE-D-25-16758Evaluating the Effectiveness of Integrating Biofeedback in the Treatment of Aggressive Outbursts (BRET-IA2): A Study ProtocolPLOS ONE

Dear Dr. Molina-Cantero,

Thank you for submitting your manuscript to PLOS ONE. After careful consideration, we feel that it has merit but does not fully meet PLOS ONE’s publication criteria as it currently stands. Therefore, we invite you to submit a revised version of the manuscript that addresses the points raised during the review process.

Please consider all the reviewers' suggestions, with emphasis on sample size selection and statistical analysis of the results.

Also, please note that including information in the introduction on: 1) how biofeedback of peripheral signals can influence CNS activity and 2) how discrimination of interoceptive signals can influence the regulation of aggressive behaviors, could contribute to a more solid justification of the study.

Depending on new results, which may be different, discussion of the results will be emphasized in the next round.

We look forward to receiving your revised manuscript.

Kind regards,

Thalía Fernández, Ph.D.

Academic Editor

PLOS ONE

Journal Requirements:

This research was funded by project Bret(IA)2, Grant PID2023-147508OB-I00 472

funded by MCIN/AEI/ 10.13039/501100011033

The authors would like to thank the participants’ parents for their cooperation in this study and the INEBIR center for allowing us to use several rooms where we will deploy the materials needed for this research. This research was funded by project Bret(IA)2 , Grant PID2023-147508OB-I00 funded by MCIN/AEI/ 10.13039/501100011033, and by FEDER, UE.

This research was funded by project Bret(IA)2, Grant PID2023-147508OB-I00 472

funded by MCIN/AEI/ 10.13039/501100011033

6. Please provide a complete Data Availability Statement in the submission form, ensuring you include all necessary access information or a reason for why you are unable to make your data freely accessible. If your research concerns only data provided within your submission, please write "All data are in the manuscript and/or supporting information files" as your Data Availability Statement.

7. Please amend your list of authors on the manuscript to ensure that each author is linked to an affiliation. Authors’ affiliations should reflect the institution where the work was done (if authors moved subsequently, you can also list the new affiliation stating “current affiliation:….” as necessary).

8. Please remove all personal information, ensure that the data shared are in accordance with participant consent, and re-upload a fully anonymized data set.

Reviewers' comments:

Reviewer's Responses to Questions

**Comments to the Author**

1. Does the manuscript provide a valid rationale for the proposed study, with clearly identified and justified research questions?

Reviewer #1: No

Reviewer #2: Yes

2. Is the protocol technically sound and planned in a manner that will lead to a meaningful outcome and allow testing the stated hypotheses?

Reviewer #1: Partly

Reviewer #2: Yes

3. Is the methodology feasible and described in sufficient detail to allow the work to be replicable?

Reviewer #1: Yes

Reviewer #2: Yes

4. Have the authors described where all data underlying the findings will be made available when the study is complete?

Reviewer #1: Yes

Reviewer #2: Yes

5. Is the manuscript presented in an intelligible fashion and written in standard English?

Reviewer #1: Yes

Reviewer #2: Yes

6. Review Comments to the Author

You may also provide optional suggestions and comments to authors that they might find helpful in planning their study.

Reviewer #1: The authors propose a research protocol aimed at evaluating the effectiveness of biofeedback in the treatment of aggressive behaviors. Additionally, as a secondary objective, they seek to establish the psychophysiological profile of aggressive episodes across different environments.

Introduction:

The introduction is well-developed; however, it lacks information on how biofeedback of peripheral signals (e.g., HRV, SCL) may influence central nervous system activity, as well as the role of interoceptive signal discrimination in the regulation of aggressive behaviors. Including this information could help provide a stronger rationale for the use of biofeedback as an intervention.

Method:

The authors calculated the sample size based on a moderate effect size, yet they do not justify why this particular effect size was chosen. Please include a meta-analysis or previous study to support this choice. Moreover, they mention that the sample size calculation was based on a t-test, which compares the means between two groups. However, they later indicate that at least four groups will be formed, making the selected statistical test inappropriate. The sample size should be recalculated using the appropriate statistical test and providing a justification for the expected effect size. Additionally, consider the effect size for both between-group comparisons and within-subject comparisons.

In the description of the sessions, it is unclear whether the evocation sessions will involve the same episode each time. Furthermore, in the biofeedback session, it is not specified whether, beyond recognizing the interoceptive sensations associated with increased sympathetic activity, participants will also be taught regulatory skills using physiological feedback.

The proposed statistical analyses (Mann-Whitney and Wilcoxon) are not adequate for comparing the four proposed groups. These tests also do not allow for testing interactions between factors. It is recommended to propose a parametric analysis and only mention the possibility of non-parametric tests in the event that data do not meet the necessary assumptions for parametric analysis.

Discussion:

The authors refer to Freud's psychoanalytic theory in the discussion section. This theory not only lacks scientific support but is also unrelated to the authors’ proposed intervention, which follows a cognitive approach.

The role of various brain structures in the regulation of aggressive behaviors is discussed, but no hypothesis is offered regarding the potential mechanism of change when training interoceptive awareness through biofeedback. Please include such a hypothesis in the discussion section.

Study Limitations:

The authors mention that:

"The application period of biofeedback therapy, along with the time required to recognize the physiological variables used to identify the preparatory state that precedes an aggressive outburst, might be too short to achieve significant results to validate the therapy."

If this is acknowledged in advance, why not include more biofeedback sessions? In fact, it would be easy to incorporate them into all sessions for the experimental group. Consider modifying the protocol to include additional biofeedback sessions

Reviewer #2: Dear Editor,

Thanks for the opportunity to submit my review report titled “Evaluating the Effectiveness of Integrating Biofeedback in the Treatment of Aggressive Outbursts (BRET-IA2): A Study Protocol”

The protocol highlights a comprehensive overview of the materials and methods required to evaluate the effectiveness of the use of biofeedback in the treatment of aggressive episodes in children and adolescents.”

The protocol is quite detailed and well thought out. The authors have provided details of all the variables that will be measured and how they will be measured during the study, including how the proposed intervention will be implemented.

Here are my comments

The sample size section is too scanty and lacks core details. What effect size was used to arrive at the final sample size of 62? What sample size estimation technique was used? Did the authors account for the non-response rate? Details must be provided

It is a bit problematic to remove an outlier based on a threshold. It is also possible that there may be genuine values that are outliers based on the threshold defined by the interquartile range specified. If you just remove these outliers, your results may be biased. Once you identify an outlier based on the interquartile range threshold, you need to investigate the source of the outlier before you can determine the appropriate solution.

Also, note that outcomes that are not normally distributed cannot be summarized with mean and standard deviations. Since this is a protocol, kindly indicate the summary measures for non-normally distributed outcomes (median and interquartile range) because it is possible that the outcome measure will not be normally distributed after data collection.

Under statistical analysis, the authors have already assumed that the residual of the outcome measure (assessment of cognitive therapy with biofeedback) will not be normally distributed. Therefore, they proposed the Mann-Whitney-Wilcoxon test, a non-parametric method, instead of the one-way ANOVA (a parametric test). Since this is just a protocol, parametric and non-parametric alternatives should be proposed. The same applies to Friedmann’s test. For study protocols, I humbly believe that presenting parametric and non-parametric approaches is the way to go, as authors can only determine the true distribution of the outcome measures when they have data.

There should be a section that summarizes the primary and secondary outcome measures and the measurement scale of these outcome measures (continuous, binary, count, ordinal, nominal etc)

How would the authors address issues of missing values or observations after the implementation of the intervention? This must be included in the protocol

7. PLOS authors have the option to publish the peer review history of their article (what does this mean?). If published, this will include your full peer review and any attached files.

Reviewer #1: **Yes: **Mauricio González-López

Reviewer #2: No

---

## [Author Response · Author response to Decision Letter 1]

6 Jun 2025

Response to Reviewers and Editor’s comments

First of all, the authors want to thank reviewers and editor for their useful comments, and for allowing the resubmission of this Study Protocol. Below is a copy of the issues detected by reviewers and editor that I will proceed to answer.

Answers to Editor’s Comments

Editor #1.1: Please ensure that your manuscript meets PLOS ONE’s style requirements, including those for file naming. The PLOS ONE style templates can be found at https://journals.plos.org/plosone/s/file?id=wjVg/PLOSOne_formatting_sample_main_body.pdf and https://journals.plos.org/plosone/s/file?id=ba62/PLOSOne_formatting_sample_title_authors_affiliations.pdf

Answer #1

We have already revised the format of the manuscript to meet the PLOS ONE’s style.

Editor #1.2: We note that the grant information you provided in the ‘Funding Information’ and ‘Financial Disclosure’ sections do not match. When you resubmit, please ensure that you provide the correct grant numbers for the awards you received for your study in the ‘Funding Information’ section.

Answer #2

Done.

Editor #1.3: Thank you for stating the following financial disclosure:

“This research was funded by project Bret(IA)2, Grant PID2023-147508OB-I00 Funded by MCIN/AEI/ 10.13039/501100011033”

Please state what role the funders took in the study. If the funders had no role, please state: “The funders had no role in study design, data collection and analysis, decision to publish, or preparation of the manuscript.” If this statement is not correct you must amend it as needed.P lease include this amended Role of Funder statement in your cover letter; we will change the online submission form on your behalf.

Answer #3:

Thank you for changing the online submission. We have included the information in the cover letter and added the role of funder statement.

Editor #1.4: Thank you for stating the following in the Acknowledgments Section of your manuscript: The authors would like to thank the participants’ parents for their cooperation in this study and the INEBIR center for allowing us to use several rooms where we will deploy the materials needed for this research. This research was funded by project Bret(IA)2 , Grant PID2023-147508OB-I00 funded by MCIN/AEI/ 10.13039/501100011033, and by FEDER, UE.

This research was funded by project Bret(IA)2, Grant PID2023-147508OB-I00 472 Funded by MCIN/AEI/ 10.13039/501100011033

Answer #4

We have fixed the grant information and added the correct text to be uploaded to the online submission in the cover letter.

Editor #1.5: Your ethics statement should only appear in the Methods section of your manuscript. If your ethics statement is written in any section besides the Methods, please move it to the Methods section and delete it from any other section. Please ensure that your ethics statement is included in your manuscript, as the ethics statement entered into the online submission form will not be published alongside your manuscript.

Answer #5

We have moved the Ethics statement to the Methods section.

Editor #1.6: Please provide a complete Data Availability Statement in the submission form, ensuring you include all necessary access information or a reason for why you are unable to make your data freely accessible. If your research concerns only data provided within your submission, please write “All data are in the manuscript and/or supporting information files” as your Data Availability Statement.

Answer #6

The manuscript describes a study protocol so there are no data available at this moment.

Editor #1.7: Please amend your list of authors on the manuscript to ensure that each author is linked to an affiliation. Authors’ affiliations should reflect the institution where the work was done (if authors moved subsequently, you can also list the new affiliation stating “current affiliation:….” As necessary).

Answer #7

All authors are linked to their affiliations.

Editor #1.8: Please remove all personal information, ensure that the data shared are in accordance with participant consent, and re-upload a fully anonymized data set.

Answer #8

There are no preliminary data in this study. We are considering creating a database after the experiment.

Editor #1.9: Please review your reference list to ensure that it is complete and correct. If you have cited papers that have been retracted, please include the rationale for doing so in the manuscript text, or remove these references and replace them with relevant current references. Any changes to the reference list should be mentioned in the rebuttal letter that accompanies your revised manuscript. If you need to cite a retracted article, indicate the article’s retracted status in the References list and also include a citation and full reference for the retraction notice

Answer #9

We have performed a thourogh revision of references to guarantee that it is complete and correct. Below are the new titles added according to reviewers’ comments:

11. Chen WG, Schloesser D, Arensdorf AM, Simmons JM, Cui C, Valentino R, et al. The emerging science of interoception: sensing, integrating, interpreting, and regulating signals within the self. Trends in neurosciences. 2021;44(1):3–16.

12. D’Andrea W, Nieves N, Van Cleave T. To thine own self be true: interoceptive accuracy and interpersonal problems. Borderline personality disorder and emotion dysregulation. 2022;9(1):6. June 6, 2025 16/20

13. Feldman MJ, MacCormack JK, Bonar AS, Lindquist KA. Interoceptive ability moderates the effect of physiological reactivity on social judgment. Emotion. 2023;23(8):2231.

25. Karjalainen S, Kujala J, Parviainen T. Neural activity is modulated by spontaneous and volitionally controlled breathing. Biological Psychology. 2025;197:109026. doi:10.1016/j.biopsycho.2025.109026.

26. Baldini A, Patron E, Gentili C, Scilingo EP, Greco A. Novel VR-Based Biofeedback Systems: A Comparison Between Heart Rate Variability- and Electrodermal Activity-Driven Approaches. IEEE Transactions on Affective Computing. 2024;doi:10.1109/TAFFC.2024.3352424.

27. Jung H, Yoo HJ, Choi P, Nashiro K, Min J, Cho C, et al. Changes in Negative Emotions Across Five Weeks of HRV Biofeedback Intervention were Mediated by Changes in Resting Heart Rate Variability. Applied Psychophysiology and Biofeedback. 2025;50:25–48. doi:10.1007/s10484-024-09674-x.

31. Cohen J. The effect size. Statistical power analysis for the behavioral sciences. Abingdon: Routledge. 1988; p. 77–83.

53. Van Buuren S, Van Buuren S. Flexible imputation of missing data. vol. 10. CRC press Boca Raton, FL; 2012

56. Lane RD, Schwartz GE. Levels of emotional awareness: acognitive-developmental theory and its application to psychopathology. The American journal of psychiatry. 1987;144(2):133–143.

Answers to Reviewers’ Comments

Reviewer #1.1: The introduction is well-developed; however, it lacks information on how biofeedback of peripheral signals (e.g., HRV, SCL) may influence central nervous system activity, as well as the role of interoceptive signal discrimination in the regulation of aggressive behaviors. Including this information could help provide a stronger rationale for the use of biofeedback as an intervention.

Answer #1.1

We would like to thank the reviewer for the suggestions. These have provided us with valuable information to enrich the introduction section. We have included some pieces of information in the introduction regarding interoception and the effect of bioffedback on peripheral signals.

Reviewer #1.2: The authors calculated the sample size based on a moderate effect size, yet they do not justify why this effect size was chosen. Please include a meta-analysis or previous study to support this choice. Moreover, they mention that the sample size calculation was based on a t-test, which compares the means between two groups. However, they later indicate that at least four groups will be formed, making the selected statistical test inappropriate. The sample size should be recalculated using the appropriate statistical test and providing a justification for the expected effect size. Additionally, consider the effect size for both between-group comparisons and within-subject comparisons.

Answer #1.2

We would like to clarify that the study includes two groups only: an intervention group and a control group. We set four subgroups during the recruitment to build balanced control and intervention groups according to gender and age. Therefore, the sample size calculation was appropriately based on a two-sample t-test, which compares means between two independent groups. We have now clarified this in the manuscript to avoid confusion.

Regarding the effect size, we selected a moderate effect size (Cohen’s d = 0.5) as it represents a practically meaningful difference in the context of clinical decision-making, even if smaller effects may also be statistically detectable in larger samples and it is a commonly used convention in behavioral and clinical research when planning exploratory studies [Cohen, 1988]. For a small effect size (Cohen’s d=0.2) we would need to recruit more than 300 participants, and we do not have human and material resources to complete this study in relativately short time.

We found a study that investigated whether pededing physiological and motion data measured by a reist-worn biosensor can predit aggression to others [Goodwin, 2019] and the study recruited only 20 young people.

[Cohen, 1988] Cohen J. The effect size. Statistical power analysis for the behavioral sciences. Abingdon: Routledge. 1988; p. 77–83.

[Goodwin, 2019] Goodwin, Matthew S., et al. "Predicting aggression to others in youth with autism using a wearable biosensor." Autism research 12.8 (2019): 1286-1296.

Reviewer #1.3: In the description of the sessions, it is unclear whether the evocation sessions will involve the same episode each time. Furthermore, in the biofeedback session, it is not specified whether, beyond recognizing the interoceptive sensations associated with increased sympathetic activity, participants will also be taught regulatory skills using physiological feedback.

Answer #1.3:

We have cleared that the use of different videos to ellicit an outburst at laboratory. Regarding the therapy, the manuscript explains that during block 3 both control and intervention groups apply cognitive therapy. The difference between them is the use or not of biofeedback to enhance interoception.

Reviewer #1.4: The proposed statistical analyses (Mann-Whitney and Wilcoxon) are not adequate for comparing the four proposed groups. These tests also do not allow for testing interactions between factors. It is recommended to propose a parametric analysis and only mention the possibility of non-parametric tests if data do not meet the necessary assumptions for parametric analysis.

Answer #1.4

We would like to clarify that the study does not involve four groups, but rather two conditions compared within the same participants (i.e., a repeated measures design). Therefore, the Wilcoxon signed-rank test was proposed as the primary statistical method to compare paired samples when the assumptions for parametric tests are not met. The Mann-Whitney U test was mentioned in the context of potential between-group comparisons in exploratory analyses, but it is not central to the primary analysis.

Nonetheless, in accordance with the reviewer’s suggestion, we have revised the manuscript to indicate that a parametric test will be used if the assumptions of normality and homogeneity are satisfied. The non-parametric alternative (Wilcoxon) is now presented as a contingency plan should those assumptions not hold.

We thank the reviewer for highlighting the importance of clearly specifying the statistical approach.

Reviewer #1.5: The authors refer to Freud’s psychoanalytic theory in the discussion section. This theory not only lacks scientific support but is also unrelated to the authors’ proposed intervention, which follows a cognitive approach.

Answer #1.5

Thank you for your insightful observation. We acknowledge that the mention of Freud’s psychoanalytic theory in the discussion section may not be directly relevant to the cognitive framework underpinning our intervention.

In light of this, we have removed the reference to psychoanalytic theory to maintain conceptual and methodological consistency with the cognitive approach adopted in our study. We appreciate your feedback in helping us ensure a clearer and more focused discussion.

Reviewer #1.6: The role of various brain structures in the regulation of aggressive behaviors is discussed, but no hypothesis is offered regarding the potential mechanism of change when training interoceptive awareness through biofeedback. Please include such a hypothesis in the discussion section.

Answer #1.6

Thank you for this important observation. We agree that providing a plausible hypothesis regarding the mechanism of change would strengthen the discussion. We have now added a theoretical explanation suggesting that interoceptive awareness training via biofeedback may modulate activity in brain regions involved in emotion regulation—particularly the insula, anterior cingulate cortex, and prefrontal areas—which in turn can reduce dysregulated aggressive responses. These structures are known to support both interoception and top-down control over emotional and behavioral impulses. The revised discussion includes this hypothesis with appropriate references.

Reviewer #1.7: If this is acknowledged in advance, why not include more biofeedback sessions? In fact, it would be easy to incorporate them into all sessions for the experimental group. Consider modifying the protocol to include additional biofeedback sessions.

Answer #1.7

We acknowledge the concern regarding the duration of biofeedback exposure and its potential impact on therapeutic outcomes. The decision to limit the number of biofeedback sessions was based on several considerations, including time constraints, feasibility in real-world clinical settings, and the need to maintain participant engagement throughout the intervention.

While we recognize that more sessions might enhance familiarity with physiological signals, our protocol integrate biofeedback into key moments of the intervention to optimize learning and transferability. Furthermore, the sessions are supported with self-monitoring tools to reinforce interoceptive awareness beyond the formal biofeedback training.

If the results indicate insufficient exposure, we will consider protocol adjustments in future trials. For the current study, however, we believe the structure keeps a balance between scientific rigor and practical applicability.

Reviewer #2.1: The sa

---

## [Decision Letter · Decision Letter 1]

16 Jun 2025

Evaluating the Effectiveness of Integrating Biofeedback in the Treatment of Aggressive Outbursts (BRET-IA2): A Study Protocol

PONE-D-25-16758R1

Dear Dr. Molina-Cantero,

We’re pleased to inform you that your manuscript has been judged scientifically suitable for publication and will be formally accepted for publication once it meets all outstanding technical requirements.

Kind regards,

Thalía Fernández, Ph.D.

Academic Editor

PLOS ONE

Additional Editor Comments (optional):

Reviewers' comments:

Reviewer's Responses to Questions

**Comments to the Author**

1. Does the manuscript provide a valid rationale for the proposed study, with clearly identified and justified research questions?

Reviewer #2: Yes

2. Is the protocol technically sound and planned in a manner that will lead to a meaningful outcome and allow testing the stated hypotheses?

Reviewer #2: Yes

3. Is the methodology feasible and described in sufficient detail to allow the work to be replicable?

Reviewer #2: Yes

4. Have the authors described where all data underlying the findings will be made available when the study is complete?

Reviewer #2: Yes

5. Is the manuscript presented in an intelligible fashion and written in standard English?

Reviewer #2: Yes

6. Review Comments to the Author

You may also provide optional suggestions and comments to authors that they might find helpful in planning their study.

Reviewer #2: The authors have addressed all my previous comments. The content of the manuscript has improved significantly

7. PLOS authors have the option to publish the peer review history of their article (what does this mean?). If published, this will include your full peer review and any attached files.

Reviewer #2: No

---

## [Editor Report · Acceptance letter]

PONE-D-25-16758R1

PLOS ONE

Dear Dr. Molina-Cantero,

I'm pleased to inform you that your manuscript has been deemed suitable for publication in PLOS ONE. Congratulations! Your manuscript is now being handed over to our production team.

Kind regards,

on behalf of

Dr. Thalía Fernández

Academic Editor

PLOS ONE